# Lifestyle habits associated with elevated depressive symptoms among healthcare workers during the COVID-19 pandemic

**Keitaro Takahashi**[1], **Narimasa Katsuta**[1,2*], **Hiroshi Fukuda**[2,3], **Mizue Saita**[2,3], **Tetsutaro Nagaoka**[2,4], **Wataru Urasaki**[5,6], **Shuko Nojiri**[5,7], **Satoshi Hori**[2], **Toshio Naito**[2,3], **Tadafumi Kato**[1]

**1** Department of Psychiatry, Faculty of Medicine, Juntendo University, Tokyo, Japan, **2** Department of Safety and Health Promotion, Juntendo University, Tokyo, Japan, **3** Department of General Medicine, Faculty of Medicine, Juntendo University, Tokyo, Japan, **4** Department of Respiratory Medicine, Juntendo University, Tokyo, Japan, **5** Clinical Research and Trial Center, Juntendo University, Tokyo, Japan, **6** Department of Information Sciences, Tokyo University of Science, Noda City, Chiba, Japan, **7** Medical Technology Innovation Center, Juntendo University, Tokyo, Japan

\* nkatsuta@juntendo.ac.jp

## Abstract

This study sought to explore the relationship between lifestyle changes among healthcare workers during the COVID-19 pandemic and positivity on the Center for Epidemiologic Studies-Depression Scale (CES-D). We also hypothesized that physical inactivity which may have occurred under such behavioral restrictions would have had a particularly strong impact on CES-D positivity, and verified this hypothesis. This observational cohort study was carried out as a component of the mandatory health checkup program for employees at Juntendo University Hospital in Tokyo, Japan.in 2022. All 4,786 employees included in the study. Among them, 4700 valid responses and 4436 consents to the study were obtained. The prevalence of CES-D in 2022 was 32.5%, higher than that in 2019 before the COVID-19 pandemic (27.5%). Moreover, we summarized the 2022 employee variables for each CES-D result (negative vs. positive group). A notable difference was identified between the two groups concerning. age, sex, BMI, occupation, living alone, total working hours, and sleep duration. Logistic regression analysis was performed using the binary category as the outcome variable (i.e., CES-D positive findings). The factors related to positive CES-D that were not present in 2019 but only in 2022 were items such as "To smoke cigarettes" and "Less than 4,000 steps." We obtained results that support our hypothesis, such as the possibility that the impact of the "new lifestyle" on step count could affect depressive symptoms. Additionally, we were able to obtain findings that suggest a relationship between the pandemic's impact and depressive symptoms. When restrictions on movement are relaxed, immediate rebuilding of better lifestyle habits is necessary to maintain good mental health among healthcare workers.

**Data availability statement:** Research data are not shared. The raw data belonged to the present study cannot be made publicly available because the disclosure of personal data was not included in the research protocol of the present study. The data are not publicly available due to privacy and ethical restrictions. The contact point for accessing the data is the Research Ethics Committee Faculty of Medicine, Juntendo University (email: hongo-rinri@juntendo.ac.jp).

**Funding:** The authors received no specific funding for this work.

**Competing interests:** The authors have declared that no competing interests exist.

## 1. Introduction

Coronavirus disease 2019 (COVID-19) is a highly contagious viral infection caused by a novel zoonotic coronavirus, severe acute respiratory syndrome coronavirus 2 (SARS-CoV-2), which first appeared in Wuhan, China in December 2019 and quickly spread globally [1]. The WHO classified it as a public health emergency of international concern (PHEIC) on January 31, 2020, and subsequently declared it a pandemic on March 11, 2020 [2]. In May 2023, the WHO announced the end of the PHEIC declaration [3] and, in principle, basic infection control measures were no longer imposed in Japan [4]. During such a crisis, frontline healthcare workers involved in diagnosing, treating, and caring for COVID-19 patients face a heightened risk of psychological distress and mental health issues, including depression. A meta-analysis indicates that factors such as the rising number of confirmed and suspected cases, excessive workloads, shortages of personal protective equipment, extensive media coverage, limited availability of certain medications, and perceptions of insufficient support can exacerbate the mental strain on these workers [5]. Additionally, their resilience may be undermined by abrupt and often unstable changes in their work environments and practices [5].

Given that numerous studies have documented an increase in depression rates among healthcare workers during the COVID-19 pandemic [6–8], we conducted a retrospective cohort study from July to August 2020 among employees at Juntendo University Hospital in Tokyo, Japan, to validate these findings. A total of 4,239 participants completed the self-assessment using the Center for Epidemiologic Studies-Depression Scale (CES-D) [9], with a score of 16 or higher considered indicative of depression. According to a study that presented pooled estimates of accuracy indices at selected cutoff points of the CES-D scale using recommended hierarchical meta-analytic methods, the CES-D demonstrates acceptable screening accuracy for detecting depression [10].

In 2020, 31.3% of employees tested positive for depression, marking the highest rate observed in the past decade. Among the different occupations, nurses had the highest positivity rate (43.2%), and the statistically significant relationship was observed between younger age and CES-D positivity ($P < .0001$) [11].

While medication and psychotherapy are the first choice of treatment for depression, in recent years, concepts, such as "lifestyle medicine," have been proposed [12]. The concept of "lifestyle medicine" is expected to treat and prevent depression by promoting physical activity and exercise [13]; enhancing diet [14], proper rest, and sleep [15]; reducing recreational substances, such as nicotine, drugs, and alcohol [15,16]; and improving social interaction [17], among other things.

In Japan, the "new lifestyle [18]" (i.e., (1) basic infection control measures for individuals, (2) basic lifestyle in daily life, (3) lifestyle for each situation, and (4) new style of work style) had been recommended during the COVID-19 pandemic. Based on this "new lifestyle," citizens (especially healthcare workers) had been imposed strict restrictions on movement, eating out, exercising in closed spaces like gyms, and attending public events.

Based on this background, we thought that the significant changes in people's lifestyles due to COVID-19 may have made healthcare workers, who were at the

forefront of COVID-19 treatment, more susceptible to depression. While previous studies in countries such as the United States and China have demonstrated an increase in depressive symptoms during the COVID-19 pandemic using the CES-D scale, targeting the general public [19,20], and studies of healthcare workers have also shown similar trends, but several reports [21,22] are limited to certain occupations such as residents or nurses, and there are few studies [23] that investigated many occupations such as doctors, nurses, physical therapists, and clerks. Additionally, our institution has conducted similar research examining depressive symptoms during the pandemic [11]. However, there has been no study that specifically identifies which lifestyle changes, known risk factors for depression, are particularly important within the context of lifestyle changes brought about by the pandemic. Furthermore, to the best of our knowledge, there are few studies that have investigated CES-D scores within the same population before and after the pandemic.

From this perspective, we believe it is highly meaningful to contribute to the existing literature by incorporating a new focus on "lifestyle" changes and conducting a study that compares the pre- and post-pandemic periods within the same population over time. This approach adds a valuable new perspective to the body of work and enriches our understanding of the impact of lifestyle changes on depressive symptoms during the pandemic.

We also hypothesized that physical inactivity which may have occurred under such behavioral restrictions would have had a particularly strong impact on CES-D positivity.

Therefore, we tested this hypothesis by comparing the CES-D results and lifestyle habits of healthcare workers before and after the COVID-19 pandemic.

## 2. Methods

### 2.1. Ethics statement

This study adhered to the "Ethical Guidelines for Medical Research for Humans" and the Declaration of Helsinki. The research protocol was approved by the Juntendo University School of Medicine Ethics Committee (approval no. E22-0392).

As a retrospective study utilizing medical records collected after informed consent was obtained (approval no. E21-0356), an opt-out approach was implemented. Details about the study design were made available on the Juntendo University Hospital website, and all participants were given the option to opt out of the study.

The date of data access for this study was 10/4/2023, and all data was anonymized.

### 2.2. Study design

This study is an observational cohort study that extracted the results of a web-based health checkup conducted as part of a mandatory health screening at Juntendo University Hospital in Tokyo, Japan, in 2022. The data was collected on April 10, 2023.

The target population consisted of all employees working at Juntendo University Hospital (a total of 4,786 individuals), including physicians, residents, nurses, pharmacists, clinical laboratory technicians, administrative staff, as well as other healthcare-related professionals such as nursing assistants, dietitians, psychologists, and social workers.

Regarding the sample size, of the 4,786 individuals who participated in the health checkup, valid responses were obtained from 4,700, and 4,436 individuals consented to participate in the study. No opt-out requests were recorded. Ultimately, data from the 4,436 participants were analyzed. Additionally, longitudinal analysis was conducted for 2,504 individuals who had comparable data from both 2019 and 2022.

As for the type of sampling, a census survey was conducted involving all employees of Juntendo University Hospital, which corresponds to a convenience sampling method. This survey was not voluntary but was conducted as part of a regular health checkup.

The eligibility criteria were staff who underwent the health checkup at Juntendo University Hospital in 2022, provided valid responses to the questionnaire, and consented to participate in the study (n = 4,436). The exclusion criteria included individuals who left questions unanswered, provided incomplete responses, or opted out (none in this study).

The CES-D was employed for self-assessment, with a score of ≥ 16 being regarded as indicative of depression. Statistical analyses were carried out using SPSS Statistics 29 (IBM Corp.). The chi-square test was applied to evaluate differences in the frequency of patient characteristics (e.g., gender). Clinical variables were compared using the two-tailed Mann–Whitney U test for two groups, and the Kruskal–Wallis test for three or more groups. A two-tailed P value of <.05 was deemed statistically significant for all tests. The relationship between participants' variables was analyzed using univariate analysis, which focused only on variables related to CES-D scores. To explore the associations between various clinical factors and depression, logistic regression analysis was performed with multiple categories as the outcome variables (i.e., CES-D positive results).Additionally, to examine the influence of each questionnaire result on the CES-D, a logistic regression analysis was conducted, considering the influence of the individual characteristics of the participants, namely 1) gender, 2) age, 3) occupation, 4) BMI, and 5) living alone/not. The statistical methods used in this study, including significance testing (the chi-square test, the two-tailed Mann–Whitney U test, and the Kruskal–Wallis test) and logistic regression analysis, were employed solely to visualize and quantitatively assess the association between two variables. They were not used to investigate any causal relationships.

## 2.3. Lifestyle questionnaire

This questionnaire was created at our facility and has been used every year for health checkups. The lifestyle questionnaire was developed for general purposes, not specifically for depression. Thus, some irrelevant questions are also included. In this study, however, we did not judge each item's relevance to depressive symptoms in the viewpoint of causal relationship and used all items for the analysis. We simply examined numerical relationship of each score and CES-D, without considering their causal relationship.

This is designed according to the items recommended by the Japanese Ministry of Health, Labour and Welfare for health checkups in the workplace [24].

We collected the following information through a structured questionnaire conducted during routine occupational health check-ups:

(1) living situation (e.g., living alone or with family),

(2) medical history (presence or absence of past illnesses),

(3) current health conditions (e.g., hypertension, diabetes, cardiovascular diseases),

(4) family history of lifestyle-related diseases (e.g., hypertension, diabetes, cardiovascular diseases),

(5) subjective symptoms (e.g., headache, chest pain),

(6) lifestyle-related factors including smoking, presence of back pain or stress, dietary and exercise habits, time spent on visual display terminal (VDT) work, sleep duration, and information processing ability,

(7) items based on Japan's Specific Health Checkups, such as adequacy of sleep, alcohol consumption, eating speed, and walking speed.

We provided details of the questions (S1).

## 3. Results

### 3.1. Comparison of 2022 and 2019 CES-D results

The CES-D prevalence in 2022 was 32.5%, which was higher than that in 2019, before the COVID-19 pandemic (27.5%). Comparing the results of all employees in 2022 and 2019, there were more females, the average age increased, and the CES-D positivity rate among nurses, researchers, and clerks worsened in 2022 (Table 1). During the COVID-19

**Table 1. Comparison of the CES-D results of the Juntendo University Hospital employees (Tokyo, Japan) in 2022 and 2019.**

| | Outcome of CES-D | | |
| | 2022 | 2019 | |
| | N = 4436 | N = 4443 | P value |
|---|---|---|---|
| Sex M/F | 1621/2815 | 1715/2728 | **.048** |
| Age (Y) | 38.3 ± 12.3 | 37.7 ± 12.1 | **.022** |
| Number and the positive rate of CES-D by occupation | | | |
| Doctor | 1098 (21.7%) | 1273 (18.4%) | .051 |
| Nurse | 1271 (43.9%) | 1299 (40%) | **.047** |
| Medical technologist | 247 (33.2%) | 232 (34.5%) | .841 |
| Researcher | 771 (27.8%) | 632 (18.4%) | **<.001** |
| Resident | 74 (23%) | 98 (19.4%) | .702 |
| Clerk | 597 (32.3%) | 513 (26.7%) | **.048** |
| Pharmacist | 114 (37.7%) | 120 (25.8%) | .07 |
| Others | 264 (36.7%) | 276 (29%) | .068 |
| Average score by occupation | | | |
| Doctor | 9 (5–14) | 6 (2–13) | **<.001** |
| Nurse | 14 (9–21) | 13 (6–20) | **<.001** |
| Medical technologist | 11 (7–18) | 11.5 (5–19) | .577 |
| Researcher | 10 (5–16) | 7 (2–13) | **<.001** |
| Resident | 10 (5.25–14.75) | 6.5 (2–12) | **.034** |
| Clerk | 12 (6–17) | 9 (4–16) | **<.001** |
| Pharmacist | 11.5 (7–20) | 9 (4–16) | **.01** |
| Others | 12 (7–20) | 10 (5.75–17) | **.008** |

Note: Bold type is used to denote statistically significant P values.

Abbreviation: CES-D, the Center for Epidemiologic Studies Depression Scale.

pandemic, the average CES-D scores worsened for all occupations except for medical technologists. Moreover, we summarized the 2022 employee variables for each CES-D result (negative vs. positive group). A notable difference was found between the two groups in relation to age, sex, BMI, occupation, living alone, total working hours, and sleep duration (Table 2). We also provided details of the results of the "lifestyle" items (S1).

### 3.2. Logistic regression analysis associated with CES-D positivity in 2019

A logistic regression model was used to calculate the adjusted odds ratios with 95% confidence intervals for factors associated with CES-D positivity. For all employees in 2019 (total: 4,443), this analysis was performed using the binary category as the outcome variable (i.e., CES-D positive findings in 2019) to investigate the factors associated with CES-D positivity before the COVID-19 pandemic. Considering the influence of the participants' individual characteristics: 1) gender, 2) age, 3) occupation, 4) BMI, and 5) living alone/not, the questionnaire results (single regression, correlation with a P value of <.2) were selected as dependent variables. A forest plot was created for items that were significantly associated (P < .05) (Fig 1). Furthermore, the factors associated with CES-D positivity only in 2019 but not in 2022 were: "Q30 (To gain 10 kg since you were 20)" and "Q35 (Amount of alcohol consumed)."

### 3.3. Logistic regression analysis associated with CES-D positivity in 2022

We also analyzed the data of employees in 2022 using the same method as the logistic regression analysis for 2019, as described above. For all employees in 2022 (total: 4,436), this analysis was performed using the binary category as the

**Table 2. CES-D results for the Juntendo University Hospital employees (Tokyo, Japan) in 2022.**

| Classification | Total | Outcome of CES-D | | p |
| --- | --- | --- | --- | --- |
| | | Negative group | Positive group | |
| | N=4436 | N=2994 | N=1442 | |
| Variables | | | | |
| Age | 38.3±12.3 | 39.4±12.8 | 36.0±11.1 | **<.001** |
| Sex, M/F | 1621/2815 | 1220/1774 | 401/1041 | **<.001** |
| BMI | 22.0±3.6 | 22.1±3.5 | 21.9±3.6 | **.002** |
| Occupation | | | | |
| Doctor | 1098 | 860 (78.3%) | 238 (21.7%) | **<.001** |
| Nurse | 1271 | 713 (56.1%) | 558 (43.9%) | |
| Medical technologist | 247 | 165 (66.8%) | 82 (33.2%) | |
| Researcher | 771 | 557 (72.2%) | 214 (27.8%) | |
| Resident | 74 | 57 (77.0%) | 17 (23.0%) | |
| Clerk | 597 | 404 (67.7%) | 193 (32.3%) | |
| Pharmacist | 114 | 71 (62.3%) | 43 (37.7%) | |
| Others | 264 | 167 (63.3%) | 97 (36.7%) | |
| Living alone/not | 1392/ 3044 | 813/2181 | 579/863 | **<.001** |
| Total working hours/day | 8.9±2.1 | 8.8±2.1 | 9.1±2.2 | **<.001** |
| Time of VDT/day | 4.7±2.7 | 4.6±2.7 | 4.8±2.7 | .116 |
| Time of sleeping/day | 6.1±1.0 | 6.2±1.0 | 5.9±1.1 | **<.001** |

Note: Bold type is used to denote statistically significant P values.

Abbreviation: CES-D, the Center for Epidemiologic Studies Depression Scale.

outcome variable (i.e., CES-D positive findings in 2022) to determine the factors associated with CES-D positivity during the COVID-19 pandemic. Similar factors (single regression, correlation with a P value of <.2) were selected as dependent variables. A forest plot was created for items that were significantly associated (P<.05) (Fig 2). The factors related to CES-D positivity that were not present in 2019, but only in 2022, were:"Q3 (To smoke cigarettes)" and "Q18 (Less than 4,000 steps)."

### 3.4. Logistic regression analysis for CES-D positive conversions from 2019 to 2022

Subsequently, we analyzed the factors expected to worsen the mental state of employees during the COVID-19 pandemic. For the employees in 2022 who also had paired data of negative CES-D for those in 2019 (total 1,872), logistic regression analysis was performed using binary category as the outcome variable (i.e., CES-D positive findings in 2022), and similar factors (single regression, correlation with a P value of <.2) were selected as the dependent variables. A forest plot was created for items that were significantly associated (P<.05) (Fig 3). Questions related to "Q5 (Stress control)," "Q4 (To feel stressed)" and "Q36 (To sleep well)," had the highest odds ratios.

## 4. Discussion

### 4.1. Characteristics of healthcare workers and mental health

In Japan, the number of COVID-19 patients has surged in January and August 2021 and July 2022. During this time, most medical institutions were extremely busy and healthcare workers were usually under mental stress. The Juntendo University Hospital is a leading healthcare facility that has treated critical COVID-19 patients due to its capability to offer advanced medical care. Therefore, it was presumed that the CES-D positivity rate was high (32.5%) because the study

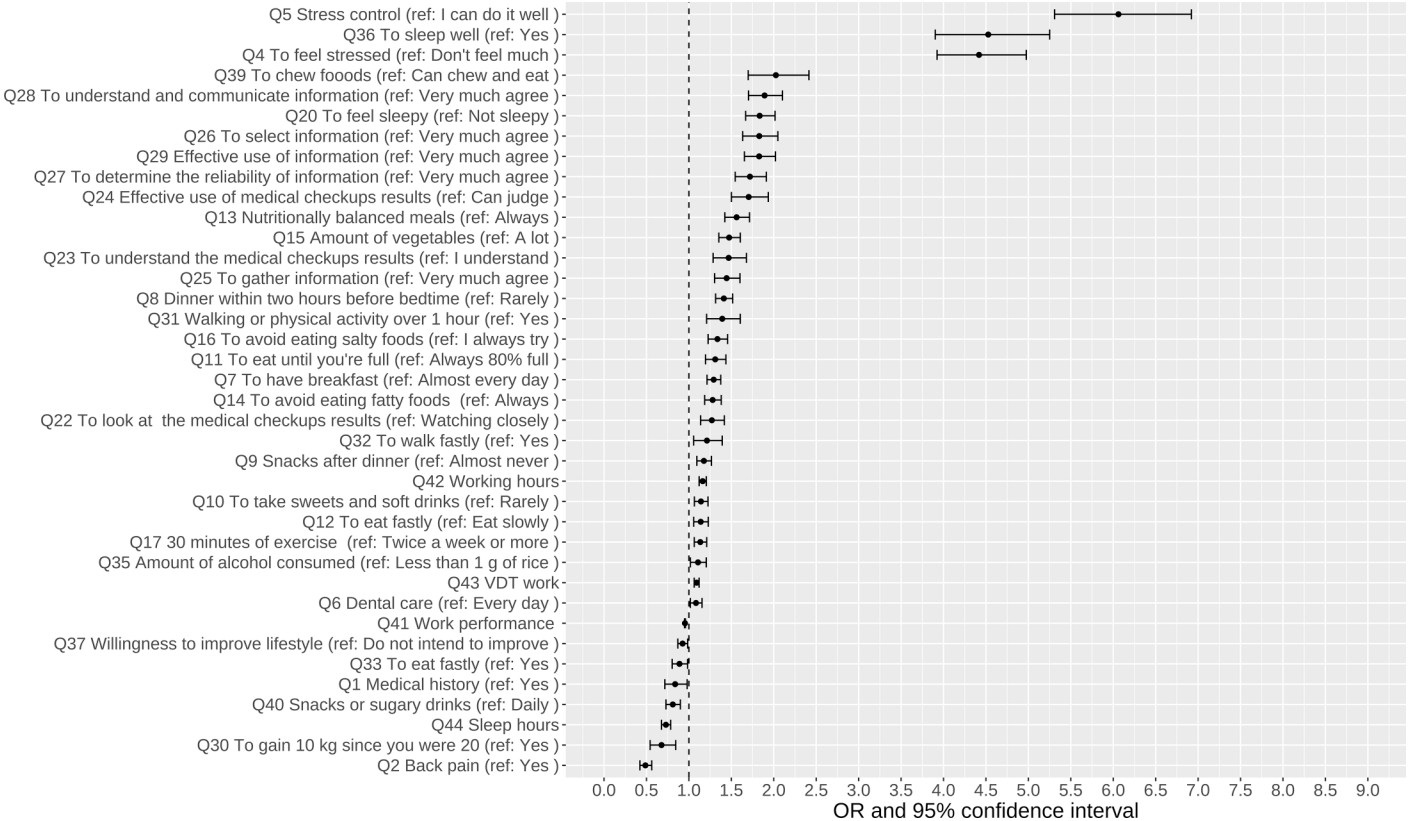

**Fig 1. Results of the logistic regression analysis associated with CES-D positivity in 2019.** Abbreviation: CES-D, the Center for Epidemiologic Studies Depression Scale.

period coincided with the COVID-19 epidemic. However, few studies have assessed the mental health of healthcare workers using the CES-D scale. In the Japanese general hospital (848 participants), the CES-D positivity rate was 27.9% [25], whereas in the Iranian urban hospital (140 participants), the CES-D positivity rate was 57.9% [26].

In the present study, younger age and female gender influenced the prevalence of depression, and this result was similar to our previous study, which was conducted at the same institution in 2020 [11]. In a previous review and meta-analysis encompassing 13 cross-sectional investigations during the COVID-19 pandemic that included a total of 33,062 healthcare workers, women and nurses were found to be at a higher risk for depression than others [27]. Moreover, multiple reports had indicated that young female healthcare workers are more susceptible to adverse psychological effects during the COVID-19 pandemic [5,28–30]. The occupation with the highest CES-D positivity rate in 2019 and 2022 was nursing. This result is similar to reports from other facilities, reaffirming the importance of mental care for nurses.

In terms of occupation in Japan, the high proportion of women among nurses is considered to be one of the factors contributing to the higher prevalence of depressive symptoms observed in women. This may be largely attributable to biological sex differences, as the prevalence of depression is generally higher in women than in men. Additionally, because nurses frequently interact with patients, they may experience greater psychological burden during outbreaks of emerging infectious diseases. Furthermore, in clinical settings where future prospects are uncertain, nurses often have no choice but to follow physicians' instructions. Under pandemic conditions, this lack of autonomy in their duties may also contribute to the worsening of depressive symptoms [31]. In a report on ICU nurses, it is stated that experiences of "distance from

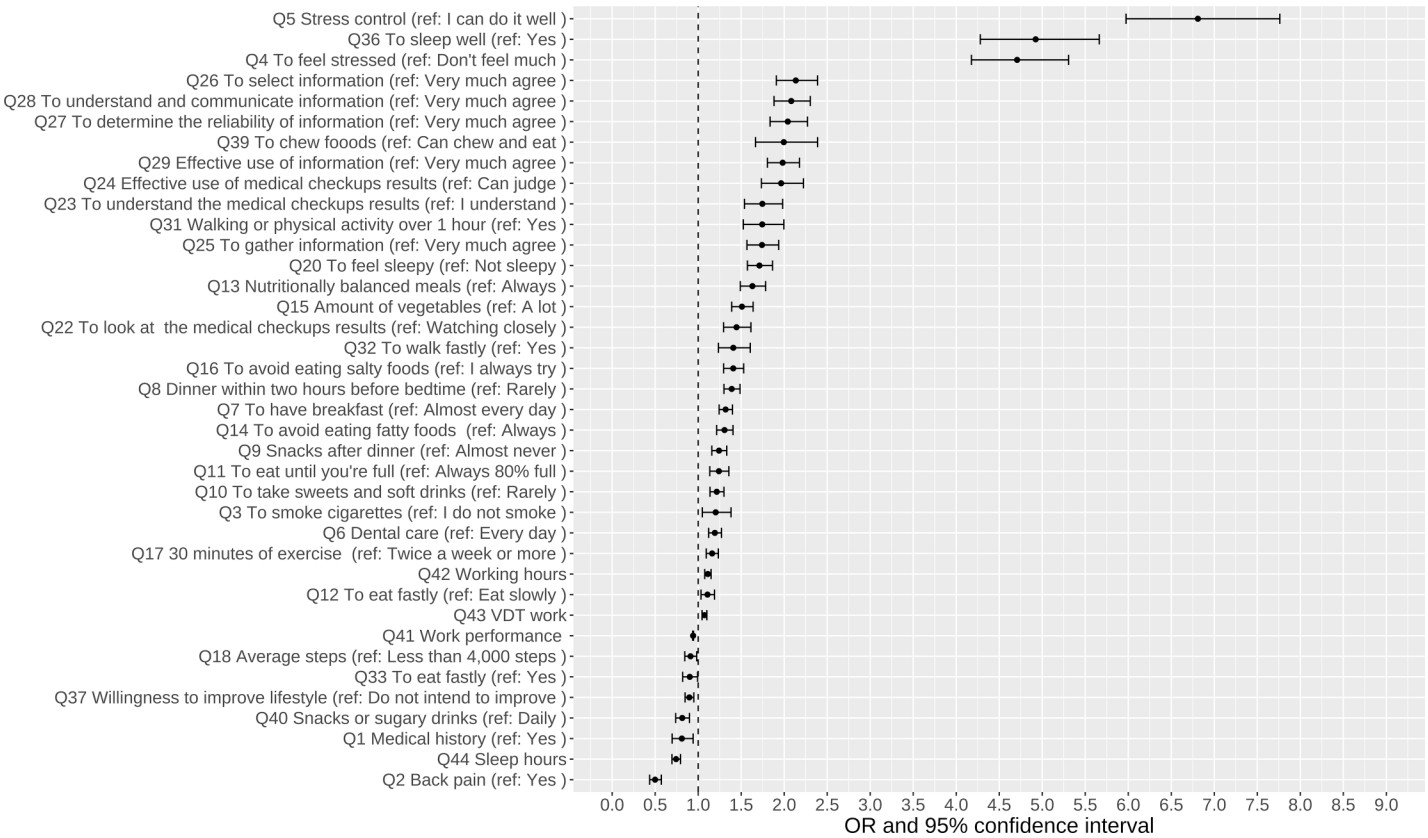

**Fig 2. Results of the logistic regression analysis associated with CES-D positivity in 2022.** Abbreviation: CES-D, the Center for Epidemiologic Studies Depression Scale.

holistic nursing" (the difficulty in maintaining human connections with patients due to infection control measures, issues with the allocation of nursing care, and problems in specialized care) and dissatisfaction with "organizational inefficiency" (lack of support for nurses, issues with the supply of resources, and inefficiencies in management) were reported. These factors may have contributed to psychological stress, which in turn could have influenced the development of depressive symptoms [32].

Additionally, a report on nurses in Iran during the COVID-19 pandemic highlights the importance of employment stability in providing high-quality nursing care, emphasizing the need to advocate for the prevention of short-term and unstable employment contracts. The report suggests that instability in economic aspects, such as employment contracts, may affect depressive symptoms [33].

## 4.2. Lifestyle and mental health

From 2020 to 2022, many people in Japan have taken strong voluntary preventive actions, such as avoiding going out and avoiding face-to-face conversations [34]. Therefore, lifestyle changes due to preventive behaviors lead to deterioration of mental health [35]. Moreover, a large-scale Swedish study also indicated that negative changes in lifestyle habits affected mental health during the COVID-19 pandemic [36]. In studies that use the CES-D, like our research, a previous study of American adolescents indicated that interruption of physical activity is a risk factor for depression [37]. However, to the

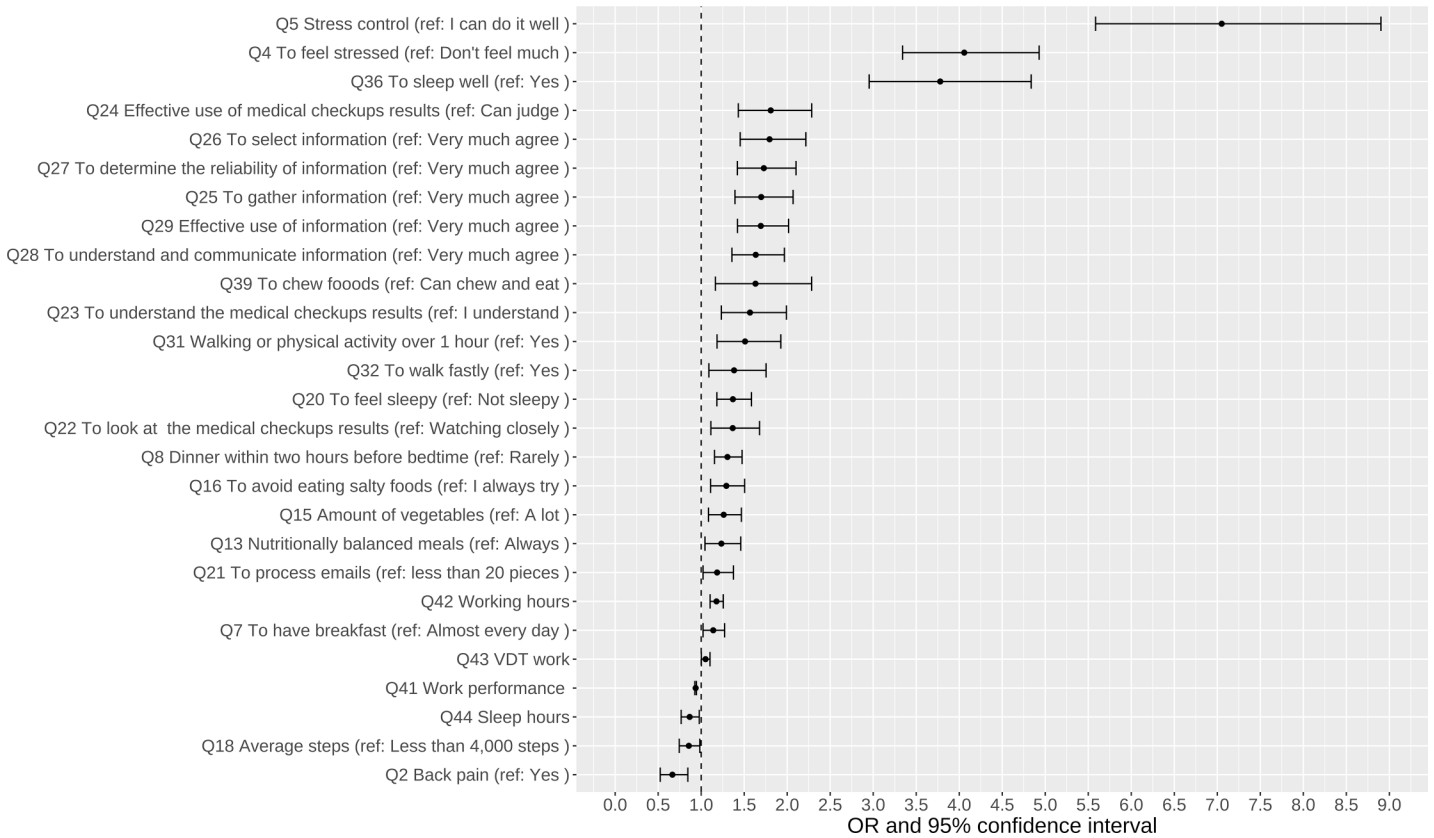

**Fig 3. Results of the logistic regression analysis for CES-D positive conversions from 2019 to 2022.** Abbreviation: CES-D, the Center for Epidemiologic Studies Depression Scale.

best of our knowledge, no studies have yet investigated the relationship between comprehensive lifestyle habits, including dietary habits, and the mental health of healthcare workers.

In this survey, the questionnaire results associated with the CES-D positivity in 2019 were "Q30 (To gain 10 kg since you were 20)" and "Q35 (Amount of alcohol consumed)" and in 2022 were "Q3 (To smoke cigarettes)" and "Q18 (Less than 4,000 steps)". In other words, the outcomes in 2019 and 2022 were largely similar. Thus, lifestyle habits for maintaining good mental health in the post-COVID-19 pandemic future would likely not change significantly. Although it was well-known that the risk of obesity caused by poor diet and increased alcohol consumption has increased globally during the COVID-19 pandemic [38], the strong association of alcohol and obesity with mental health has been even before the COVID-19 pandemic.

Because "Q18 (Less than 4,000 steps)" was associated with the CES-D positivity in 2022, it should be necessary for all Japanese citizens to make an effort to avoid reducing the number of steps taken daily even if all citizens were forced to adopt the "new lifestyle." Many studies have shown that exercise has an antidepressant effect, and physical inactivity is known to increase the risk of depressive symptoms and anxiety- [12]. As measures to address the lack of physical activity, we consider enhancing employee benefits to promote exercise, such as providing fitness facilities, and creating opportunities for physical activity during work, such as introducing standing desks.

Moreover, increasing the workers' leisure and activity time (e.g., shopping, yoga, and strength training) could help maintain their mental health [39]. Thus, by 2022, most healthcare workers were not able to live the lifestyle they desired,

which might have influenced the high CES-D positivity rate and lifestyle changes. Although it was known that physical activity and healthy eating habits are associated with lower odds of depression [14,40], effective guidance had not been proposed. Therefore, the results of this study could be useful in developing mental health programs for workers. In health coaching situations, it should be desirable to use questions related to "Q5 (Stress control)," "Q36 (To sleep well)," and "Q4 (To feel stressed)," which had the highest odds ratios in 2019 and 2022 based on interviews with employees. Additionally, regarding "Q3 (To smoke cigarettes)," which was associated with the CES-D positive in 2022, the association between smoking and the increased risk of COVID-19 deaths [41] was well known in Japan. Thus, smokers might have been more susceptible to stress during the COVID-19 pandemic. And it is generally known that smokers are more likely to develop depressive symptoms compared to non-smokers [41]. While nicotine may provide short-term stress relief, in the long term, it can affect the regulation of neurotransmitters such as serotonin and dopamine, potentially worsening mental health.

Looking at factors associated with CES-D positive conversions from 2019 to 2022, the above-mentioned Q5, Q36, and Q4 were ranked among the top ten with the highest odds ratios, and six of the remaining seven were questions related to health information (Q24–Q29) (Fig 3). Regarding information-related topics, the spread of social media had been remarkable in recent years, and several reports related to social media have been published. A true and healthy social media environment could improve mental health [42], but the spread of dangerous misinformation undermines trust and mental health, especially among young people [43]. Some reports have also indicated that increased exposure to media coverage related to COVID-19 increases anxiety [44] and depression, and that using social media during shelter-in-place or lockdown could be detrimental to the individual's emotional and mental health [45]. Among medical professionals, several occupations and employees are not well versed in their medical knowledge. Thus, staff members who are confused about any health information on social media should be educated on this matter.

In summary, the findings of this study suggest that lifestyle changes during the COVID-19 pandemic, particularly physical inactivity, smoking, and increased exposure to information through social media, contributed to depressive symptoms among healthcare workers. These results align with the study's objective of examining how these factors influenced CES-D positivity and support the initial hypothesis. When restrictions on movement have been relaxed, immediate rebuilding of better lifestyle habits is necessary to maintain good mental health.

### 4.3. Work and mental health

Although it was not ranked highly among factors associated with the CES-D positive conversions from 2019 to 2022 (Fig 3), significant differences were also observed in working hours and VDT work hours. Several studies have investigated the association between long working hours and mental health. Females who worked long hours (41 hours or more per week) were more likely to develop depression than those who worked normal hours (odds ratio: 2.2). In contrast, for men, no relationship was observed between long working hours and depression [46]. Moreover, a study targeting white-collar workers in Japan indicated that those who worked 50 hours or more per week had a higher risk for developing mental illness (odds ratio: 1.36) [47]. Possible measures to improve long working hours in the workplace include improving work efficiency and proper task distribution, reviewing working hours (such as introducing a flexible work schedule), and strengthening mental health support (such as implementing stress checks and providing mental health training).

Owing to the increased proportion of contactless work and telecommuting during the COVID-19 pandemic, workers' VDT working hours increased significantly [48]. In our study, we also assessed VDT work time using logistic regression analysis, which is rarely reported, and found that not only working hours but also VDT work time was significantly related to CES-D positive conversions from 2019 to 2022. As VDT work has become more common over the past 3 years, including the encouragement of active email communication and the introduction of online medical consultations and online meetings, controlling VDT work hours might also be helpful in improving the mental health of employees.

The limitations of this study include its observational design and dependence on a single depression scale (CES-D). For instance, factors related to occupational stress and other psychological assessments were not fully explored.

Additionally, since the lifestyle questionnaire was self-reported, the responses may have been influenced by the workers' personal biases. The lifestyle questionnaire used in this study has not been formally validated.

## Supporting information

**S1 Fig. Details of lifestyle questions 1–44.**
(ZIP)

## Acknowledgments

The authors thank Enago for carefully proofreading the manuscript. The authors would like to express their deepest appreciation to Ms. Kanami Ito for collecting and organizing the data.

## Author contributions

**Conceptualization:** Narimasa Katsuta, Hiroshi Fukuda, Mizue Saita, Tetsutaro Nagaoka, Satoshi Hori, Toshio Naito.

**Data curation:** Narimasa Katsuta, Hiroshi Fukuda, Mizue Saita, Tetsutaro Nagaoka, Satoshi Hori, Toshio Naito.

**Formal analysis:** Narimasa Katsuta, Wataru Urasaki, Shuko Nojiri.

**Project administration:** Hiroshi Fukuda, Tetsutaro Nagaoka.

**Supervision:** Toshio Naito, Tadafumi Kato.

**Writing – original draft:** Narimasa Katsuta, Keitaro Takahashi.

**Writing – review & editing:** Tadafumi Kato.

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
