## [Decision Letter · Decision Letter 0]

24 Mar 2025

PMEN-D-24-00524

Lifestyle habits associated with elevated depressive symptoms among healthcare workers during the COVID-19 pandemic

PLOS Mental Health

Dear Dr. Katsuta,

Thank you for submitting your manuscript to PLOS Mental Health. After careful consideration, we feel that it has merit but does not fully meet PLOS Mental Health’s publication criteria as it currently stands. Therefore, we invite you to submit a revised version of the manuscript that addresses the points raised during the review process.

The reviewers have provided their comments below on your submission. Please reply with a point by point response. We draw your attention specifically to the concerns raised surrounding the reporting of your methodology. It is a requirement of publication in PLOS Mental Health that experiments, statistics, and other analyses are performed to a high technical standard and are described in sufficient detail (https://journals.plos.org/mentalhealth/s/criteria-for-publication#loc-3). 

We note that one of the reviewers has requested the citation of specific works. As always, we recommend that you please review and evaluate the requested works to determine whether they are relevant and should be cited. It is not a requirement to cite these works. We appreciate your attention to this request.

We look forward to receiving your revised manuscript.

Kind regards,

Joanna Tindall, PhD

Staff Editor

PLOS Mental Health

Journal Requirements:

1. Please provide additional details regarding participant consent. If you are reporting a retrospective study of medical records or archived samples, please ensure that you have discussed whether all data were fully anonymized before you accessed them and/or whether the IRB or ethics committee waived the requirement for informed consent. If patients provided informed written consent to have data from their medical records used in research, please include this information.

- https://doi.org/10.1002/npr2.12217

- DOI: 10.1002/npr2.12217

- https://doi.org/10.3389/fpubh.2023.1131971

In your revision ensure you cite all your sources (including your own works), and quote or rephrase any duplicated text outside the methods section. Further consideration is dependent on these concerns being addressed.

Additional Editor Comments (if provided):

Comments from PLOS Editorial Office: We note that one or more reviewers has recommended that you cite specific previously published works. As always, we recommend that you please review and evaluate the requested works to determine whether they are relevant and should be cited. It is not a requirement to cite these works. We appreciate your attention to this request.

Reviewers' comments:

Reviewer's Responses to Questions

**Comments to the Author**

1. Does this manuscript meet PLOS Mental Health’s publication criteria ? Is the manuscript technically sound, and do the data support the conclusions? The manuscript must describe methodologically and ethically rigorous research with conclusions that are appropriately drawn based on the data presented.

Reviewer #1: Yes

Reviewer #2: No

2. Has the statistical analysis been performed appropriately and rigorously?

Reviewer #1: Yes

Reviewer #2: No

3. Have the authors made all data underlying the findings in their manuscript fully available (please refer to the Data Availability Statement at the start of the manuscript PDF file)?

Reviewer #1: Yes

Reviewer #2: No

4. Is the manuscript presented in an intelligible fashion and written in standard English?

Reviewer #1: Yes

Reviewer #2: No

5. Review Comments to the Author

Reviewer #1: Dear authors

I would like to thank you for giving me the opportunity to review the manuscript entitled “Lifestyle habits associated with elevated depressive symptoms among healthcare workers during the COVID-19 pandemic”. The manuscript presents an interesting and potentially valuable study. However, it should be carefully revised. Please find my comments:

- Introduction: While the introduction highlights the pandemic's impact, it could better integrate findings from existing global studies to situate the research within the broader literature. For instance, comparisons with international CES-D studies would provide deeper context.

- The hypothesis could be more explicit regarding which specific lifestyle habits (e.g., physical inactivity, smoking) were expected to associate with CES-D positivity.

- Method: The description of the lifestyle questionnaire lacks specificity. Supplementary figures are referenced but should also be briefly described in the main text.

- Clarify how questions were validated for relevance to depressive symptoms.

- Results: Highlight key logistic regression findings directly in the text rather than relying heavily on figures.

- The manuscript notes a higher prevalence of depressive symptoms among females. Discuss this finding in more detail, including potential occupational or societal influences.

- Discussion: Expand the discussion on how specific factors like smoking and physical inactivity (e.g., <4,000 steps) contribute to mental health disparities. Are these findings consistent with pre-pandemic data?

- Discuss other issues can be influence depressive symptoms among healthcare workers. The following article can be used:

1. https://www.frontiersin.org/journals/public-health/articles/10.3389/fpubh.2022.1034624/full

2. https://www.mdpi.com/1660-4601/18/23/12548

- Provide more actionable recommendations for addressing CES-D positivity based on the findings. For example, what workplace changes could mitigate factors like high working hours or lack of physical activity?

Reviewer #2: The manuscript, "Lifestyle habits associated with elevated depressive symptoms among healthcare workers during the COVID-19 pandemic" does not meet the editorial criteria for publication. Although this is original research, the overall similarity percentage is higher than expected (26%) according to the Turnitin software report. It is observed that there is no logical coherence between the title, aim, results and research design, it is stated that it is a retrospective study and in other parts of the manuscript it is stated that an online survey was used, which corresponds to a prospective research. Similarly, it is not specified which participants were followed over time to see who did or did not develop the outcome of interest. The validity and reliability of the measurement instruments were not specified, which may affect the quality of the research, as the results may be inconsistent or erratic. On the other hand, the results are not consistent with the purpose of the study. The statistical processing of the data is not consistent with the objective of the study, significant differences have been made, but the objective of the study indicates that it is the association of two variables. In the abstract the conclusion has been omitted; but it has not been omitted in the discussion. The manuscript is generally understandable and written in standard English, but there are syntactic and grammatical errors that should be improved to make it clearer and more precise (inconsistency in verb tenses, some sentences are too long or lack appropriate connectors, affecting fluency, some prepositions can be improved). The ethical principles applied in the research are specified in lines 112 to 119. The manuscript does not comply with the main STROBE guidelines (in addition to what is described, it is noted that the methodological aspects are incomplete; the population, sample size, type of sampling, eligibility criteria, selection methods, follow-up methods,are not stated). In the discussion the results are not presented according to the study objective, the discussion is not made of the key results of the association of the two study variables (lifestyles and depressive symptoms). In addition, no data available at the beginning of the PDF file of the manuscript are included, only inclusion of figures is mentioned.

6. PLOS authors have the option to publish the peer review history of their article (what does this mean? ). If published, this will include your full peer review and any attached files.

**Do you want your identity to be public for this peer review?** For information about this choice, including consent withdrawal, please see our Privacy Policy .

Reviewer #1: No

Reviewer #2: No

---

## [Decision Letter · Decision Letter 1]

11 Jul 2025

Lifestyle habits associated with elevated depressive symptoms among healthcare workers during the COVID-19 pandemic

PMEN-D-24-00524R1

Dear Dr. Katsuta,

We are pleased to inform you that your manuscript 'Lifestyle habits associated with elevated depressive symptoms among healthcare workers during the COVID-19 pandemic' has been provisionally accepted for publication in PLOS Mental Health.

Best regards,

Karli Montague-Cardoso

Staff Editor

PLOS Mental Health

Reviewer Comments (if any, and for reference):

Reviewer's Responses to Questions

**Comments to the Author**

1. If the authors have adequately addressed your comments raised in a previous round of review and you feel that this manuscript is now acceptable for publication, you may indicate that here to bypass the “Comments to the Author” section, enter your conflict of interest statement in the “Confidential to Editor” section, and submit your "Accept" recommendation.

Reviewer #1: All comments have been addressed

Reviewer #2: All comments have been addressed

2. Does this manuscript meet PLOS Mental Health’s publication criteria ? Is the manuscript technically sound, and do the data support the conclusions? The manuscript must describe methodologically and ethically rigorous research with conclusions that are appropriately drawn based on the data presented.

Reviewer #1: Yes

Reviewer #2: Yes

3. Has the statistical analysis been performed appropriately and rigorously?

Reviewer #1: Yes

Reviewer #2: Yes

4. Have the authors made all data underlying the findings in their manuscript fully available (please refer to the Data Availability Statement at the start of the manuscript PDF file)?

Reviewer #1: Yes

Reviewer #2: Yes

5. Is the manuscript presented in an intelligible fashion and written in standard English?

Reviewer #1: Yes

Reviewer #2: Yes

6. Review Comments to the Author

Reviewer #1: Thank you to the authors for addressing my comments in the revised version of the manuscript.

However, I noticed that the initially submitted file (instead of the clean version) appears to have been used, and the revised file with tracked changes is currently included in the proof. I kindly request that the correct version — the revised manuscript — be used for publication to avoid any confusion or errors.

Thank you for your attention to this matter.

Best regards,

Reviewer #2: Having reviewed the corrected manuscript and the supporting responses, I am satisfied that the comments made in the initial review have been adequately addressed.

7. PLOS authors have the option to publish the peer review history of their article (what does this mean? ). If published, this will include your full peer review and any attached files.

**Do you want your identity to be public for this peer review?** For information about this choice, including consent withdrawal, please see our Privacy Policy .

Reviewer #1: No

Reviewer #2: No
